# Antimicrobial Resistance in Bacterial Strains of Agricultural Interest: Predictions Based on Genomic Data

**DOI:** 10.3390/antibiotics15010014

**Published:** 2025-12-20

**Authors:** Eloísa Pajuelo, Manuel Medina-Rodríguez, Noris J. Flores-Duarte, Bouchra Doukkali, Jennifer Mesa-Marín, Ignacio D. Rodríguez-Llorente, Salvadora Navarro-Torre

**Affiliations:** Department of Microbiology and Parasitology, Faculty of Pharmacy, University of Sevilla, 41012 Sevilla, Spain; b02merom@uco.es (M.M.-R.); nflores@us.es (N.J.F.-D.); doukkali@us.es (B.D.); mesam@us.es (J.M.-M.); irodri@us.es (I.D.R.-L.); snavarro1@us.es (S.N.-T.)

**Keywords:** antimicrobial resistance (AMR), plant growth promoting bacteria (PGPB), minimal inhibitory concentration (MIC), genomics, genome mining, PATRIC, CARD

## Abstract

**Background**: Plant growth promoting bacteria (PGPB) are non-pathogenic bacteria that enhance plant growth through several mechanisms such as nutrient mobilization, phytohormones production, defense against phytopathogens, and alleviation of plant stress. Hence, these bacteria are used as ecologic biofertilizers to diminish the use of agrochemicals. Nevertheless, some PGPR strains can harbor antibiotic resistance determinants and the possibility of spreading them upon releasing these bacteria is an environmental concern. **Objectives:** The objectives of this work are as follows: (1) evaluating the antibiotic resistance in a collection of PGPB, and (2) prospecting antibiotic resistance genes in the genomes of PGPB in order to predict the risk for antibiotic resistance dissemination. **Methods**: The resistance towards 12 antibiotics in a collection of 20 PGPB (10 Gram-positive and 10 Gram-negative strains) has been evaluated using disk diffusion in agar, broth microdilution, and agar dilution tests. In addition, the whole genomes of six strains have been sequenced in order to find the correlation between the resistance levels and AMR genes by using bioinformatic tools. **Results**: The results indicated a wide range of halo diameters, but in general Gram-negatives showed higher resistance compared to Gram-positives. The four most resistant strains and the two more susceptible strains were selected for further analysis and sequencing the whole genomes. The resistant strains were identified as *Achromobacter spanius* N6, *Leclercia adecarboxylata* H17, *Priestia aryabhattai* strain MHA1, and *Bacillus cereus* N25. The susceptible strains were identified as *Pantoea* sp. S3 and *Priestia megaterium* MS4. Mining antibiotic resistance genes in the genomes confirmed the existence of resistance determinants responsible for the phenotypic behavior, indicating the potential of genomics for predicting antibiotic resistance in PGPB. However, there was not an exact correspondence between the presence of the genes and the level of resistance, suggesting the existence of additional regulatory mechanisms. **Conclusions**: The information obtained by genomics must be complemented experimentally by tests for antibiotic resistance determination. In this regard, it is necessary to develop a global antibiotic resistance database for PGPB, due to the difficulty of interpretation of the antibiotic susceptibility tests after comparing the experimental results with those tabulated for clinical species.

## 1. Introduction

Antibiotic resistance in bacteria, particularly pathogenic species, represents one of the greatest challenges humans will face in the coming decades. Its effects are already becoming evident through the reduced efficacy of traditional antibiotics and, consequently, a decreasing availability of effective treatments, coupled with increasing difficulty in discovering new antibiotics [1]. Additionally, it is estimated that in 2021, approximately 4.71 million people died due to antimicrobial resistance (AMR), and that by 2050, this number could rise to around 8.71 million deaths per year [2].

Bacteria have developed a wide range of mechanisms to avoid the action of antibiotics, including efflux pumps, enzymatic inactivation, target modification, sequestration/chemical modification of the antibiotics, reduced permeability, biofilm formation, etc. [3,4,5]. The coexistence of several of these mechanisms within a single genome account for the multidrug resistance profiles commonly observed in clinical settings.

Although resistance can emerge through mutation, its global dissemination primarily relies on horizontal gene transfer (HGT). Plasmid-mediated conjugation is the dominant pathway: a meta-analysis of 113 studies reported transfer frequencies spanning 12 orders of magnitude, with peaks in manure-fertilized agricultural soils and livestock effluents [6]. In biofilms, cell proximity and the extracellular matrix facilitate the conjugation even of non-conjugative plasmids, via co-integrated plasmids or auxiliary systems [7]. Natural transformation and bacteriophage-mediated transduction also contribute to gene exchange, although with a more restricted taxonomic scope. These processes give rise to a genetic mosaic that links environmental, animal, and human reservoirs, driven by co-selection involving antimicrobials, heavy metals, and biocides. The spread of broad-host-range plasmids, such as IncP-1, poses an emerging threat under the One Health framework [8].

PGPB (plant growth promoting bacteria) enhance plant development through both direct and indirect mechanisms, being considered as ecological biotechnological tools [9,10]. Direct mechanisms include nutrient solubilization, phytohormone production, and the stimulation of lateral root formation to improve nutrient uptake [11]. Indirectly, they help mitigate abiotic stress and exert biocontrol over phytopathogens [12]. Genera such as *Bacillus*, *Pseudomonas*, *Azospirillum*, and *Rhizobium* include non-pathogenic PGPB strains frequently used as natural biofertilizers [13,14,15]. However, the environmental release of such bacterial inoculants may present a risk if they act as potential vectors of antibiotic resistance genes (ARGs). A meta-analysis of 125 studies showed that around 85% of phosphate-solubilizing bacteria isolated from soil harbored at least one ARG, with enzymatic deactivation (48.7%) and active efflux (25.2%) as the most prevalent resistance mechanisms [16]. Recent examples include *Serratia rubidaea* ED1, isolated from agricultural soil and resistant to seven antibiotics with multiple efflux pumps, and *Delftia tsuruhatensis* [17]. The use of commercial bioinoculants (which may contain 10^8^–10^9^ CFU/mL) represents a potential risk for horizontal ARG transfer within agricultural ecosystems [16]. For this reason, international entities advocate the implementation of phenotypic and genomic screening prior to the approval of such microbial fertilizers.

Standardized methods for determining antibiotic resistance (as established by institutions such as CLSI in the US or EUCAST in Europe) rely on comparing experimental results to breakpoint tables that define whether a strain is considered susceptible or resistant to a given antibiotic. However, these tables (published and updated yearly) are based on data from clinical isolates. Therefore, most of the time, when testing antibiotic susceptibility in agricultural strains, there is no dedicated reference database, since most antibiotics lack established breakpoints for beneficial soil bacteria such as PGPB. Consequently, susceptibility results are interpreted using criteria defined for clinically relevant species, which may lead to misinterpretation. Moreover, EU Regulation 2019/1009 outlines the requirements for microbial biostimulants, such as a minimum viable concentration and the absence of specific pathogens, but does not address antimicrobial resistance profiles, leaving a regulatory gap. This situation hinders comparison across studies, limits the quality control of biofertilizers, and constitutes a barrier to their safe integration into sustainable agricultural systems.

Genomic analysis has become a key tool for identifying and contextualizing resistance genes in environmental bacteria [18]. Databases such as PATRIC (Pathosystems Resource Integration Center) provide access to thousands of fully annotated bacterial genomes, along with tools for comparative analysis, mobile element visualization, and the identification of resistance and virulence factors [19]. Likewise, the CARD (Comprehensive Antibiotic Resistance Database) combines highly curated annotations with protein homology-based prediction models, facilitating the detection of both known and emerging resistance genes [20].

Our department holds a big collection of PGPB strains which have been isolated during the past 10 years, including both Gram-positive and Gram-negative bacteria with good plant growth promoting activities. However, the antibiotic resistance profile of these microorganisms has not been evaluated. In this context, the present work aims to (a) determine the antibiotic susceptibility profiles of 20 selected PGPB based on experimental testing; (b) identify resistance genes within their genomes using experimental databases and sequence similarity analyses; and (c) establish potential correlations with the experimental data obtained in order to help to predict the risk of disseminating antimicrobial resistance determinants upon releasing particular PGPB.

## 2. Results

### 2.1. Collection of Bacterial Strains and Antibiotics Selected for the Study

In the department of Microbiology and Parasitology of the Faculty of Pharmacy, there was already a pre-existing collection of PGPB strains isolated from agricultural soils. Ten Gram-positive strains and ten Gram-negative strains were selected for this study (Table 1). The selected organisms were those with properties considered optimal for promoting plant growth as previously published [21,22,23,24,25]. The strains were retrieved from frozen cultures stored at −80 °C in glycerol and plated on TSA Petri dishes, followed by incubation at 28 °C for 24 h.

The antibiotics used in this study are those recommended by the French regulation on biofertilizers ANSES (2020) [26] as the current Spanish and European legislations do not include microbial resistance criteria in the regulation of biofertilizers. These antibiotics are listed in Table 2.

### 2.2. Determination of Antimicrobial Susceptibility by the Disk Diffusion Method on Agar

In the first step, the susceptibility was tested by the agar diffusion test (Appendix A). Data of halo diameters for the selected strains are shown in Table 3 (Gram-negatives) and Table 4 (Gram-positives). Results for ampicillin revealed fully resistant strains (both within Gram-negative and Gram-positive collections) giving no inhibition halo (ϕ = 0). Among the susceptible strains, the halo diameters in Gram-negatives were in general lower (9–14 mm) than in Gram-positive (13–17 mm), indicating the higher ampicillin resistance in the Gram-negatives. For gentamicin, no strain gave full resistance. The halo diameters for Gram-negative (18–26 mm) and Gram-positive strains (17–30 mm) were comparable, although slightly lower for Gram-negatives. Regarding kanamycin, the strain N30 was resistant. The rest of both collections gave diameter halos comprised between 11 and 25 mm for Gram-negatives, and 14–33 mm for Gram-positives, indicating a higher susceptibility of Gram-positives. For streptomycin, the Gram-negative strain H17 was resistant (ϕ = 0). Again, the halo diameters of the Gram-negative collection were in general lower (11–22 mm) than those of Gram-positive (9–27), indicating the higher resistance of Gram-negatives. For tetracycline, the strains H4 and N30 (both Gram-negatives) did not produce any inhibition halo, and were resistant. In the Gram-positive collection, all strains gave inhibition halos with ϕ values between 8 and 25 mm, indicating high variability in the level of resistance, probably due to different resistance mechanisms and/or gene doses of genes associated with antibiotic resistance. From all these data, the first conclusion is that in general the Gram-negative strains showed a higher resistance to several antibiotics as compared to Gram-positives.

Furthermore, there were antibiotics specifically used for Gram-negatives: Ciprofloxacin, colistin, and Fosfomycin. Two strains, H17 and N1, showed resistance against Ciprofloxacin, whereas the rest displayed halos comprised between 8 and 32 mm; H17 and N3 were resistant towards colistin (ϕ = 0), and the rest produced halos between 10 and 16 mm. For Fosfomycin, N1 and N6 were fully resistant (ϕ = 0), and the rest formed halos with ϕ between 9 and 37 mm, reflecting high variability.

Regarding the Gram-positives, Chloramphenicol, Clindamycin, Erythromycin, and Vancomycin were the specific antibiotics used. For Chloramphenicol, MHA1 showed full resistance (ϕ = 0), and the rest ranged between ϕ17 and 29 mm, showing variable sensibility. As for Clindamycin, ϕ ranged between 6 and 31 mm (0 mm for MHA1). Erythromycin sensibility was prevalent, with ϕ being 12–39 mm, once again excluding the full resistance of MHA1. Finally, most strains were resistant to Vancomycin, with a ϕ of 7–24 mm, except for MHA1, N18, and N32, which did not develop inhibition halos.

Figure 1 shows a box plot diagram of the halo diameters for the five antibiotics assayed in both the Gram-negative and the Gram-positive collections, namely, Ampicillin, Gentamicin, Kanamycin, Streptomycin, and Tetracycline. As we can see, this representation reinforces the previous conclusion: the median value of the halo diameter was lower for Gram-negative strains, indicating the higher resistance of Gram-negative bacteria. On the other hand, there were antibiotics for which the diameter halos showed very low variability (for instance, gentamicin and colistin in Gram-negatives), whereas for other antibiotics such as Ciprofloxacin, Erythromycin, Fosfomycin, and Vancomycin, large variability was found. These results could suggest the existence of multiple resistance mechanisms, an effect of gene dosage or regulatory mechanisms controlling the expression of the resistance determinants.

### 2.3. Selection of Six Strains for Further Studies

In the next step, we have determined the minimal inhibitory concentration (MIC) to antibiotics of six selected strains, for which we picked three strains among the Gram-positives and three from the Gram-negatives, selecting the most resistant and sensitive strains. Since the sensibility or the resistance depended on every specific antibiotic, we needed a criterion to select the “most sensitive strains” and “the most resistant strains” regarding the whole set of antibiotics. Among Gram-negatives, N6 and H17 were selected, as they were the most antibiotic-resistant bacteria when considering the complete spectrum. As for the most sensitive strains, S3 was selected over N3, as its halo diameters were larger. Among the Gram-positive bacteria, MHA1 was clearly the most resistant strain, followed by N25, and both strains were selected. In contrast, the most susceptible strains were MH6, followed by MR4, MS5, and MS4. However, MS4 was selected for the study over the other candidates due to the difficulty of growing MH6. Finally, the selected strains were N6, H17, S3, N25, MHA1, and MS4.

### 2.4. Determination of Minimal Inhibitory Concentration

The determination of the MIC was carried out by the microdilution test for all the antibiotics, except for Fosfomycin. For this particular antibiotic, the EUCAST (European Committee on Antimicrobial Susceptibility Testing, https://www.eucast.org/, accessed on 19 November 2025) recommends the agar dilution test in the presence of 25 mM glucose-6-phosphate [27].

The main problem at the time of assigning the categories of resistant/sensitive has been the lack of information for the species found in agriculture. It should be taken into account that breakpoints are tabulated for clinical strains, not for strains isolated from agricultural or environmental samples. In fact, the tables published yearly by both the Clinical Laboratory Standard Institute (CLSI) and EUCAST are elaborated for clinically relevant species, and it is not possible to find data for species such as *Leclercia adecarboxilata* or *Achromobacter spanius*. In this regard, the interpretation of the results to classify the strains into the resistant (R) or susceptible (S) categories has been performed following the breakpoints for *Bacillus* spp. for Gram-positives, and for *Pseudomonas* and Enterobacteriaceae for Gram-negatives [27]. When no data were available, the EUCAST guidance “When there are no breakpoints in breakpoint tables” was followed [28].

The data of MIC for MHA1 were comprised between 128 and 512 mg L^−1^, indicating the high resistance towards many antibiotics (Table 5). By contrast, the values of MIC for MS4 were very low, indicating the susceptibility towards all antibiotics but ampicillin. By its part, N25 showed high MIC for ampicillin and clindamycin and was susceptible or showed an intermediate susceptibility towards the rest of the antibiotics (Table 6). Among Gram-negatives, the strain showing the highest values of MIC was N6, whereas the most susceptible strain was S3 (Table 6). N6 showed high MIC for all antibiotics, except for ciprofloxacin and kanamycin. The resistance towards streptomycin could not be compared with any tabulated data. *Leclercia adecarboxylata* H17 showed high values of MIC for ampicillin and Fosfomycin, and was susceptible to gentamicin and kanamycin. Concerning S3, it was susceptible to most antibiotics, except for ampicillin and ciprofloxacin.

Figure 2 shows a heatmap depicting the correlation between halo diameters and the MIC. The correlations were good in general, with values between 1 and 0.7 for most cases. In general, we can see that there are similar correlation indices in Gram-positive and Gram-negative strains. For Gram-positives, the lowest correlation was found for N25 in tetracycline (0.47) and MS4 for ampicillin and clindamycin (0.57 in both cases). In Gram-negatives, the lowest correlation was found for N6 in tetracycline (0.20).

In our study, we have multiple variables, such as 20 different bacteria (10 Gram-negative and 10 Gram-positive) and 5 common antibiotics for both types of bacteria, together with specific antibiotics, determination of halo diameters, and determination of MIC by two different methods (microdilution and agar dilution). In this regard, the multifactorial nature of the data can be simplified by reducing the number of variables to two principal components. The principal components analysis (PCA, Figure 3) indicates that both axes together accounted for 83.7% of the variability, with 64.5% of variability associated with dimension 1 and 19.2% of variability associated with dimension 2. Thus, the reduction in variables did not much affect the representativity of data and the conclusions. On the other hand, the PCA allows visual representation of the distance between samples and also to visualize how the samples can be grouped. We can see that Gram-positive strains were more disperse according to component 1, while Gram-negative strains occupied a central position in the graphic and spanned more regarding component 2. This would suggest different resistance mechanisms in both types of bacteria, since the data of MIC and halos are grouped in a different way in Gram-positives and Gram-negatives. In addition, we can observe superposition of the circles englobing both types of bacteria, which could also suggest the existence of common antimicrobial resistance mechanisms [29].

### 2.5. Whole Genome Sequences and Main Features of the Assemblies

The whole genomes of the six bacteria were sequenced and a study of the assembled genomes was conducted. The main features of the sequences are shown in Table 7. The genomes had sizes comprised between 4.9 and 6.6 Mb, without significant differences between Gram-positive and Gram-negative strains. The number of contigs oscillated between 72 and 219, being little shorter in Gram-negatives. The G + C percentages clearly reflected differences between both types (34–37% in Gram-positives and 54–64% in Gram-negatives). The number of coding sequences oscillated between 4577 and 6185. There were great differences between the number of RNA genes, being much higher in Gram-positives (127–128) than in Gram-negatives (65–97). Also, the number of rRNA and tRNA were higher in Gram-positives. Finally, there were 10–20 contigs related to plasmids in the strains MHA1 and MS4 (genus *Priestia*, as we will comment on next), whereas in N25 and in Gram-negatives, the number of contigs associated with plasmids was comprised between 0 and 3.

### 2.6. Identification of the Selected Strains

Table 8 shows the identification of the strains based on the sequence of the 16S rRNA extracted from the WGS. Among the Gram-positive strains, MS4 and MHA1 were identified as *Priestia megaterium* and *P. aryabhattai*, whereas N25 corresponded to *Bacillus cereus*. Among Gram-negatives, N6 was identified as *Achromobacter spanius*, H17 as *Leclercia adecarboxylata*, and S3 corresponded to *Pantoea* sp. In this case, the comparison of the complete rRNA16S sequence gave the highest percentage of similarity with *P. conspicua* (99.64%) and also with *P. agglomerans*. (99.64%). However, after digital DNA–DNA hybridization (dDDH) comparing the whole genome, the maximum identity (69.8%) was observed with *P. pleuroti* (isolated from the fungus *Pleurotus ostreatus*) [33]. Moreover, this low identity after the comparison of the whole genome (below 70%) allowed us to suspect that it can belong to a novel species within the genus and this is the reason for us to not assign the strain to a particular species. Further studies will be performed to confirm this possibility. For the rest of isolates, the same species was assigned upon 16S rRNA sequence and dDDH.

### 2.7. Mining Antibiotic Resistance Genes in the WGS

In order to work at the prediction level, genes for antibiotic resistance have been prospected in the whole genomes of the six bacteria using the bioinformatics tools PATRIC [19] and CARD [20]. As shown in Table 9 and Table 10, the genomes of Gram-negative bacteria contain greater numbers of resistance genes against the tested antibiotics (17–26) compared to those of Gram-positive bacteria (10–17). These results were in accordance with the experimental determinations of halos and MICs.

Among the genes common to both groups are *gyrA* and *gyrB*, which are involved in resistance to ciprofloxacin through target modification. This antibiotic inhibits DNA gyrase, an essential enzyme for DNA replication. Gram-positives and Gram-negatives also shared the determinants *s12p* and *gidB*, related to resistance to aminoglycosides such as streptomycin, which act on the ribosomal small subunit. The determinant *s10p* is related to the resistance towards tetracycline. In this case, mutations in the ribosomal proteins prevent the antibiotic from interfering with protein synthesis. Another gene found in both Gram-positives and Gram-negatives was *murA*. Mutations in this gene confer resistance towards Fosfomycin by affecting the protein MurA involved in the first steps of peptidoglycan synthesis (before the action of beta-lactamases) [35].

On the other hand, some resistance genes appear exclusively in one group. For example, the resistance towards Fosfomycin mediated by *fosA* and *fosB* was found only in Gram-positives. FosA, present in MHA1, is a class of metalloenzymes able to disrupt the epoxide ring of the Fosfomycin drug, preventing its action [36], whereas *FosB*, present in MS4, codifies an enzyme catalyzing the binding of L-cysteine or bacillithiol to Fosfomycin, thus inhibiting its bactericidal effect [37]. The strain N25 was the only Gram-positive showing genes for resistance towards chloramphenicol and vancomycin. There are correlations with the diameter of halos (17 and 8 mm, respectively, Table 3). The resistance towards chloramphenicol was represented by the gene *catA* codifying the chloramphenicol acetyl-transferase, which modifies the conformation of chloramphenicol by acetylation and prevents its binding to the ribosomal large subunit [38]. In addition, the genes *VanXY, VanA/I/Pt-type*, and *VanF/M-type*—associated with resistance to vancomycin (an antibiotic that inhibits the polymerization of peptidoglycan in the bacterial cell wall)—were only found in the Gram-positive strain N25. It is noteworthy that the determinants for vancomycin and Fosfomycin resistance are shared in the same plasmid in enterococci [39]. The strain N25 also carried the gene *BcII* conferring resistance towards ampicillin, as it was confirmed experimentally by the absence of a halo (Table 4) and the high MIC (˃512 mg L^−1^) (Table 5).

Regarding the localization of these genes, all of them were found in the chromosomes of the bacterial strains, with the exception of the gene *gidB* from MHA1 conferring resistance towards streptomycin and a complete copy of the operon *Van*, found in plasmids. Concerning the number of copies, we call attention to the resistance against ciprofloxacin (two copies of *gyrAB*) in MHA1 and MS4 and two copies of some vancomycin resistance genes in N25.

Analogously, there were genes specifically found only in the Gram-negative strains. For example, β-lactamase genes from the BlaEC family, which hydrolyze the β-lactam ring of antibiotics such as ampicillin, were detected exclusively in the genome of H17 (according to the absence of an inhibition halo (Table 3).

The most singular finding concerning Gram-negatives (Table 10) was the ubiquitous presence of multiple multidrug efflux pumps belonging to three superfamilies: the resistance–nodulation–cell division (RND) antibiotic efflux pump superfamily, the Major Facilitator Superfamily (MFS), and the ABC (ATP-binding cassette) transporters. These tripartite efflux pumps are connected to channels in the outer membrane for extruding several antibiotics and/or drugs. Concerning the RND superfamily, *AcrAB/TolC* and *TolC/OpmH* were found in the strains H17 and S3 (Enterobacteriaceae) [40]. For its part, MdtABC-TolC is a multidrug efflux system of Gram-negative bacteria, including *E. coli* and *Salmonella* [41] which were was found in H17 and S3. MdtA is a membrane fusion protein; TolC is the outer membrane channel; MdtBC forms a drug transporter. In the absence of MdtB, the MdtAC-TolC has a narrower drug specificity. All these proteins are part of RND-type efflux pumps that actively expel various classes of antibiotics (e.g., tetracyclines, macrolides, chloramphenicol) in Gram-negatives, thereby reducing their intracellular concentrations [41].

The genes *MexAB/OprM, MexCD/OprI, MexHI-OpmD,* and *MexEF-OprN* also belong to the RND superfamily and were found in N6 and S3. OprM is an outer membrane factor protein found in *Pseudomonas aeruginosa* and *Burkholderia vietnamiensis*. It is part of the MexAB-OprM, MexVW-OprM, MexXY-OprM, and the AmrAB-OprM complexes and confers resistance against aminoglycoside antibiotic, phenicol antibiotic, tetracycline antibiotic, carbapenem, fluoroquinolone antibiotic, macrolide antibiotic, penicillin beta-lactam, cephalosporin, and disinfecting agents and antiseptics [42]. According to CARD, the prevalence of this sequence in *Pseudomonas aeruginosa* is 98.47% in the chromosome and 0.58% in plasmids, whereas in the PGPB *Pseudomonas fluorescens*, the prevalence was 2.78%, not being found in plasmids (https://card.mcmaster.ca/ontology/36518*,* accessed on 11th November, 2025). In our case, it was found in the chromosome of N6, with four copies, three of them associated with *MexAB/OprM* genes and one independent copy.

Concerning the Major Facilitator Superfamily (MFS), the genes *MdfA/Cmr* codify a multidrug resistant transporter conferring resistance towards multiple antibiotics [43]. In our case, these genes were found in H17 and S3. In addition, *EmrD* codifies a multidrug transporter from the Major Facilitator Superfamily (MFS) primarily found in *Escherichia coli*. EmrD couples the efflux of amphipathic compounds with proton import across the plasma membrane. This pump confers resistance towards phenicol antibiotics (chloramphenicol), disinfecting agents and antiseptics [44].

With regard to tripartite efflux pumps that belong to the family of the ABC transporters, the genes *macAB* conferring resistance towards erythromycin [45] were detected in the strains N6 and H17.

We also found the operon *marAB* conferring resistance towards multiple antibiotics [46]. We found both genes in the strain H17 and only *marA* in S3. MarA is an activator protein encoded by the *marRAB* locus. The *mar* locus is reported to mediate resistance primarily by up-regulating the efflux of some antibiotics, disinfectants, and organic solvents via the AcrAB-TolC efflux pump and down-regulating influx through Outer Membrane Protein F (OmpF) [47].

Finally, the gene H-NS is a Histone-like nucleoid-structuring protein found in *Acinetobacter baumannii* that plays a regulatory role in antibiotic resistance. Inactivation of this gene increases the resistance towards colistin [48]. This gene was found in H17 (one chromosomal copy) and S3 (five copies; three in the chromosome and two in plasmids), but not in N6, confirming the inverse relationship between the presence of this gene and the resistance towards colistin (ϕ = 13–14 mm for S3 and H17, and ϕ = 0 in N6, Table 3). Except for the two copies of H-NS in plasmids of S3, all the rest of the genes had a chromosomal localization.

## 3. Discussion

The increasing use of plant growth promoting bacteria, such as those reviewed in this study, and despite their agronomics benefits as biofertilizers, poses a potential risk due to their capacity to contain and disseminate antibiotic resistance genes [16]. In this regard, there is an increasing number of publications calling attention to this environmental problem [49,50].

At the experimental level, we determined the diameters of both halos and MICs. in our study, we have observed, in general, a higher resistance in Gram-negative strains, visualized in the box plot diagram. Particularly high was the resistance to ampicillin for which the smaller halos were found in Gram-negatives (four strains gave no inhibition halos), followed by colistin and tetracycline. Overall, we found good correlation between halo diameters and MIC values (r between 0.7 and 1), although in some cases there were lower correlations. It is possible that the availability of the antibiotic (in liquid or solid media) or additional regulatory mechanisms could explain the lack of correlation between the two techniques. In addition, sometimes the bacteria could adopt a sessile way of life, forming biofilms, which is a mechanism of resistance and elevates the values of MIC for antibiotics since the bacteria are protected from it while immersed in the polysaccharide matrix [51].

Gram-negative strains are particularly concerning in terms of antibiotic resistance. Their intrinsic resistance is often higher due to the outer membrane acting as a permeability barrier, combined with the frequent presence of efflux pumps and the ability to acquire resistance genes via plasmids. These features make them more prone to develop multidrug resistance and complicate treatment options, as reported in various clinical and environmental studies [52]. Regarding Gram-positives, the resistance towards vancomycin was outstanding, which was present in at last three strains.

The protocols for determining both, susceptibility tests with disks and MICs, are perfectly known and established. However, one problem at the time of applying these tests to PGPB has been the lack of tabulated breakpoints for these bacteria. In this regard, it has been necessary to compare our results with general breakpoints for a particular genus, for example, *Bacillus*, or even for a complete family, for instance Enterobacteriaceae or Pseudomonadaceae. It is presumed that, in the short term, national or supranational regulations will assess the problem of AMR associated with PGPB and will establish a paradigm of antibiotics for Gram-positive and Gram-negative strains, as it occurs in France. Hence, the development of a specific database with breakpoints of halo diameters and MIC based on PGPB experimental determinations is needed.

Predicting whether PGPR contains AMR determinants is essential before being applied to agriculture. An approach used is the analysis of the whole community antibiotic resistance phenotypic profile (cenoantibiogram) [53,54,55]. In this context, in addition, the usage of omics techniques can be extremely useful for this matter, as they show directly the determinants contained in each strain [18]. In addition, the presence of resistance genes can be approached at a metagenomic level in soil DNA by shotgun sequencing without previous cultivation [56]. These studies can shed light on the prevalence of some ARG determinants, in particular microbial communities in soils, and constitute rapid tools for improving AMR surveillance, especially upon the application of machine learning-based AMR prediction models that allow the detection of ARG in mobile genetic elements [56].

Upon the isolation of PGPB, determining the antibiotic susceptibility and prospecting the relevant genes in the genomes is becoming usual. Some recent studies concerning PGPB include not only the beneficial properties of these microorganisms, but the evaluation of their antibiotic sensitivity [57,58,59]. In Gram-negative strains, efflux pumps and β-lactamases in the strain *Pantoea agglomerans* RSO7, isolated from the rhizosphere of halophytes in coastal environments, were found [25]. As for Gram-positive strains [50], resistance genes in *Bacillus* spp. from soil were identified. These findings highlight the relevance of genome mining tools for ARG detection in both Gram-negative and Gram-positive PGPR.

In the present study, across the analyzed genomes, we detected resistance determinants associated with well-established mechanisms: target modification genes (e.g., *gyrA*, *gyrB*, *S10p*, *S12p*, *MurA*); antibiotic inactivation enzymes such as β-lactamases (BlaEC) and FosB; genes conferring resistance via absence (*gidB*); regulators modulating the expression of antibiotic resistance genes (*mar A*, *marB*); transcription factors (H-NS); etc. These findings reflect the molecular diversity of resistance strategies encoded in both Gram-positive and Gram-negative PGPR and illustrate the value of comprehensive genome mining approaches to map them. In Gram-negative strains, the existence of multiple efflux pumps belonging to three superfamilies, RND, ABC transporters, and MFS, was noteworthy. Within the tripartite RND pumps are AcrAB−TolC in Enterobacteriaceae, MexAB−OprM, MexCD−OprJ, and MexXY−OprM in *Pseudomonas aeruginosa* and MtrCDE in *Neisseria gonorrhoeae* [40]. Genes AcrAB−TolC were found in H17 and S3, whereas MexAB−OprM, MexCD−OprJ, and MexXY−OprM were found in N6. The genes MdfA/Cmr and EmrD belonging to the Major Facilitator Superfamily (MFS) were also found in Enterobacteriacae H17 and S3 and the genes *macAB* encoding a tripartite efflux pump that belong to the family of the ABC transporters were found in H17. In general, the presence of multidrug efflux pumps belonging to different families (ABC transporters, RDN efflux pumps, and MFS superfamilies) in Gram-negative strains could help explain their high levels of multidrug resistance.

The critical role of the mobile genetic elements (MGEs), such as insertions sequences, transposons, cassettes/integrons, plasmids, and bacteriophages, in the dissemination of antibiotic resistance determinants via horizontal gene transfer (HGT) makes necessary the localization of genes in the genome [60,61]. In our case, all the genes but two were located in the chromosomes of the bacteria. The genes *VanA/I/Pt*-type of the Gram-positive *Priestia megaterium* N25 and two copies of the gene H-NS of the Gram-negative S3 are in plasmids. We have not detected genes in other mobile elements. In this regard, we call attention to the fact that short reads sequencing (Illumina) and assembly were performed, which limit the detection of AMR-associated MGE. Third generation sequencing platforms generating long reads like Pacific Biosciences (PacBio) and Oxford Nanopore Technologies (ONT), together with predictive machine learning models, can provide more accurate tools for linking ARGs to mobile genetic elements [56]. The low number of ARGs found in mobile elements could be a guarantee for the lower dispersion of these resistances in the environment upon using these PGPB.

Another aspect of the foremost relevance is the number of copies, ranging from one copy in most of the genes to the maximum number of five copies for H-SN in the strain S3. However, H-NS is not properly a resistance mechanism but a master regulator of the expression of other genes [48]. The strain N6 was the one with the highest number of copies of several genes including two to three copies of the multidrug efflux pumps *MexAB-OprM, MexCD-OprJ,* and *MexHI-OpmD*, as well as two copies of *MurA* conferring resistance towards Fosfomycin.

Nevertheless, a certain discrepancy is observed between the number of identified resistance genes and the actual phenotypic resistance when the susceptibility is determined experimentally. For example, the strain S3 appeared as the most susceptible strain among the Gram-negatives studied and it was classified as susceptible in most assays. However, the analysis of the genome reported a similar total number of resistance genes (54) to more resistant strains like N6 (50) or H17 (55). This suggests that other factors such as expression levels, epigenetic regulation, or synergistic effects among resistance genes may contribute to the final resistance profile. For instance, strain MHA1, which showed strong phenotypic resistance, did not possess the highest number of resistance genes. In addition, other environmental factors such as the presence of heavy metals, co-selection of different resistance mechanisms, or the biofilm/planktonic can affect the actual level of resistance as determined experimentally [18,51,62]. This highlights the need to complement genomic analysis with experimental data when evaluating candidate biofertilizer strains.

Finally, establishing a database of breakpoints for PGPB will allow us to have a scientific criterion to define the regulatory bases for safe biofertilizers in the context of a One Health perspective, trying to avoid the spreading of AMR due to the agricultural practices related to inoculation with beneficial microorganisms [63].

### Limitations of the Work and Future Perspectives

One of the main hindrances of this work, and also one of its main conclusions, is the lack of breakpoints (for both MIC and halo diameters) for environmental bacteria, which reveals the need for establishing an updated database for these species. Regarding this point, it is important to note the low availability of data concerning the antibiotic resistances of PGPB and highlight the necessity for creating a global database of antibiotic resistance in strains of agricultural or environmental application, instead of using the breakpoints for clinical species.

Another limitation is the scarce number of strains used in this work. It is necessary to increase the determination of antibiotic susceptibility at the experimental level together with WGS to find correlations between both approaches and also with the localization in MGE.

Finally, regarding the functional validation, it would be adequate to perform expression analysis and evaluate the influence of environmental conditions in the expression levels. We plan to deepen our study and perform new analyses in the future.

## 4. Materials and Methods

### 4.1. Determination of Antimicrobial Susceptibility by the Disk Diffusion Method on Agar

The procedure was carried out following CLSI guidelines [64], using 12 cm square plates containing Mueller-Hinton (MH) agar. In addition, each bacterial strain was cultured in liquid medium (TSB) for 24 h at 28 °C and 200 rpm. Plates were inoculated with bacterial suspension with optical density equivalent to MacFarland 0.5 standard (approx. 1.5 × 10^8^ CFU/mL) using sterile swabs rotating the plate during the process to ensure uniform coverage of the surface. Disks of the antibiotics (as in Table 2) were placed on the inoculated plate with sterile twicers. Plates were incubated for 24 h at 28 °C. After incubation, the diameters of the inhibition zones around each antibiotic disk were measured in millimeters using a ruler (Appendix A). The assignment of the categories (resistant, intermediate, or susceptible) was performed after consulting the diameter of the halo in the tables elaborated every year by EUCAST [27]. When no data were available for a particular antibiotic and/or bacterial genera, the criterion of the EUCAST guidance on “When there are no breakpoints in breakpoint tables” was followed [28].

### 4.2. Determination of Minimal Inhibitory Concentration by the Broth Microdilution Method

To corroborate the antibiotic susceptibility or resistance of each organism, the MIC was determined for each strain and each available antibiotic. This was carried out by the broth microdilution test [65] in 96-well microplates (8 rows × 12 columns). Liquid cultures of the microorganisms were inoculated into MH liquid medium supplemented with decreasing concentrations of antibiotics ranging from 512 µg/mL to 0.5 µg/mL. Antibiotic stock solutions were prepared at an initial concentration of 1024 µg/mL and sterilized by filtration trough a filter of 0.22 µm pore size. The first dilution (512 µg/mL) was prepared by mixing equal volumes of MH liquid medium and the stock solution of the antibiotic. Then, serial half dilutions were performed up to final concentration 0.5 µg/mL, keeping a column for MH medium without antibiotic. Each well was filled with 0.25 mL of MH containing the appropriate antibiotic concentration, along with 5 µL of bacterial culture. A row was kept without inoculation as a control for sterility of the medium. After inoculation, the plates were incubated for 24 h at 28 °C, with the lids properly sealed to prevent desiccation and contamination.

The MIC for each microorganism–antibiotic combination was determined by visually inspecting each row of wells and identifying the first clear well, which corresponded to the lowest antibiotic concentration capable of inhibiting visible bacterial growth (Appendix A).

### 4.3. Determination of the Minimal Inhibitory Concentration for Fosfomycin

Due to the fact that Fosfomycin uptake into bacterial cells requires co-transport with glucose-6-phosphate, susceptibility testing for Fosfomycin was performed using Mueller-Hinton agar medium supplemented with 25 mM glucose-6-phosphate, as recommended over the standard method used for the other antibiotics [66]. The supplemented medium was poured into 12 cm square plates, each containing decreasing concentrations of Fosfomycin, with the goal of obtaining 12 different concentrations, starting at 512 µg/mL and halving successively down to a final concentration of 0.5 µg/mL. The last plate was reserved for Mueller-Hinton medium without antibiotic.

Each series of plates was inoculated with three different bacterial strains (in triplicate). Plates were then incubated for 24 h at 28 °C, after which the maximum concentration at which each microorganism was still able to grow was determined by visual inspection (Appendix A).

### 4.4. Isolation of Genomic DNA and Whole Genome Sequencing (WGS)

For bacterial DNA sequencing, the commercial provider MicrobesNG was selected, and their proprietary workflow was followed. This protocol was based on a cell harvesting and isolation procedure, after which the samples were shipped for further processing by the company.

From a single colony of each bacterial strain, fresh TSA agar plates were inoculated and incubated at 28 °C for 24 h. After incubation, between half and two-thirds of the plate surface was scraped and suspended in tubes containing 5–10 mL of PBS buffer, until the suspension appeared visibly turbid. The optical density (OD) of each tube was then measured at 600 nm using a spectrophotometer. If necessary, the samples were diluted in cuvettes to achieve a reliable OD reading within the optimal range of 0.8 to 1.2. Once this reading was obtained, the required volume to reach an OD of 10 was extrapolated, and this volume was centrifuged to obtain a cell pellet with sufficient biomass to ensure reliable downstream sequencing.

Finally, the pellet was resuspended in an inactivation buffer provided by the sequencing company and transferred to a pre-labeled tube (also provided), after which the samples were shipped to the company’s laboratories for Illumina shotgun metagenomic sequencing.

Illumina reads for each strain genome were trimmed using Trimmomatic [67], and the quality was determined using an in-house script. Assembly was performed using SPAdes version 3.7 [68]. The assembly metrics and other basic statistics were obtained using QUAST v. 5.3.0 software [29], CRISPRCasFinder [32], and Deeplasmid version of August, 22nd, 2024 [31] software. Finally, all genomes were annotated with Prokka version 1.14.5 [30]. The whole genomes were deposited in the DDBJ/EMBL/GenBank databases with the following accession numbers: GCA_977061975 (H17), GCA_977061995 (MHA1), GCA_977061985 (MS4), GCA_977062005 (N6), GCA_977062015 (N25), and PRJEB102903 (S3).

### 4.5. Identification of the Selected Strains

The identification of the strains used in this study was carried out using the 16S rRNA gene sequence obtained by the genome sequences using the EZBioCloud web server [69]. The identification was accurate by a digital DNA–DNA hybridization (dDDH) with the most related species using the GGDC web server [70].

### 4.6. Mining Antibiotic Resistance Determinants in the Genomes

In order to obtain a summary of the antibiotic resistance determinants from every strain’s genome, we used a tool from BV-BRC (Bacterial and Viral Bioinformatics Resource Center) which annotated the genomes using the obtained paired contigs and by comparing them to those genomes stored in specialized databases for antibiotic resistance and specialty genes for species similar to those predicted before for each individual strain. Those databases include mainly PATRIC [19] and CARD [20].

### 4.7. Statistical Analyses

The determination of halo diameters was performed in triplicate. Resulting data are the means of three determinations. The values of MIC were also assessed in triplicate. When there were discrepancies between one of the three determinations, the highest value of MIC was considered. Box plots and the heatmap were elaborated using Microsoft Excel (Microsoft Corporation, Redmond (Washington), USA, 2024), the latter using normalized data, while PCA was performed by using R version 4.4.1 (R Core Team, Vienna, Austria, 2024).

## 5. Conclusions

The determination of the susceptibility to 12 antibiotics in 20 PGPB demonstrated a higher level of resistance in Gram-negatives compared to Gram-positives. Furthermore, a good correlation was found between the data of halo diameters and MIC values. Genome mining performed in the WGS of four resistant and two susceptible strains showed the existence of common and different resistance mechanisms among the Gram-positive and Gram-negative PGPB. Gram-positive PGPB harbored genes of resistance by target modification, acquiring resistance to antibiotics affecting protein synthesis and DNA replication. Gram-negative PGPB harbored an elevated number of multidrug resistance efflux pumps of different families (RDN, ABC, MFS) that expel antibiotics from the cell. In summary, genomic analysis confirms and expands the phenotypic observations, revealing both shared and specific resistance genes and mechanisms. In addition, only two of the genes were found in mobile genetic elements (plasmids), which limits the possible horizontal gene transfer. However, since the genotype-to-phenotype relationship is not always direct, further experimental validation will be essential before approving any strain as a safe microbial inoculant. In this regard, the establishment of a global database with the susceptibility of PGPB is compulsory in order to have reference data based on agricultural or environmental strains, instead of comparing the experimental data with those tabulated for clinical species.

## Figures and Tables

**Figure 1 antibiotics-15-00014-f001:**
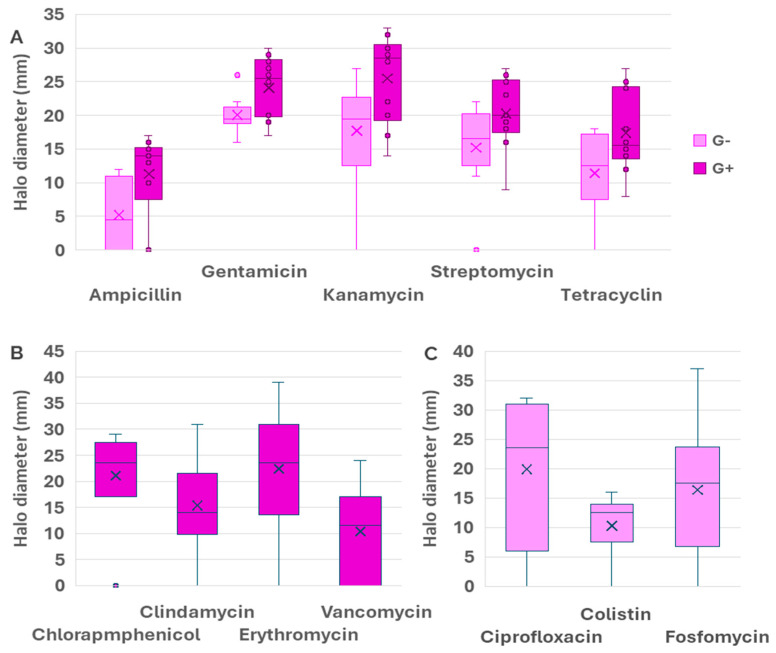
Box˗plot comparing the average values and the range intervals of halo diameters for antibiotics used in both Gram-positive and Gram-negative strains (**A**), exclusive antibiotics for Gram-positives (**B**), and exclusive antibiotics for Gram-negatives (**C**). Box indicates the quartile, and the median is indicated by a line, whilst the average is shown as a cross. Bars indicate the full range of data and circles indicate outliers, whereas crosses represent the median of the data and the horizntal lines represent the average. Data for Gram-negatives are shown in pink color whereas purple was used for Gram-positives.

**Figure 2 antibiotics-15-00014-f002:**
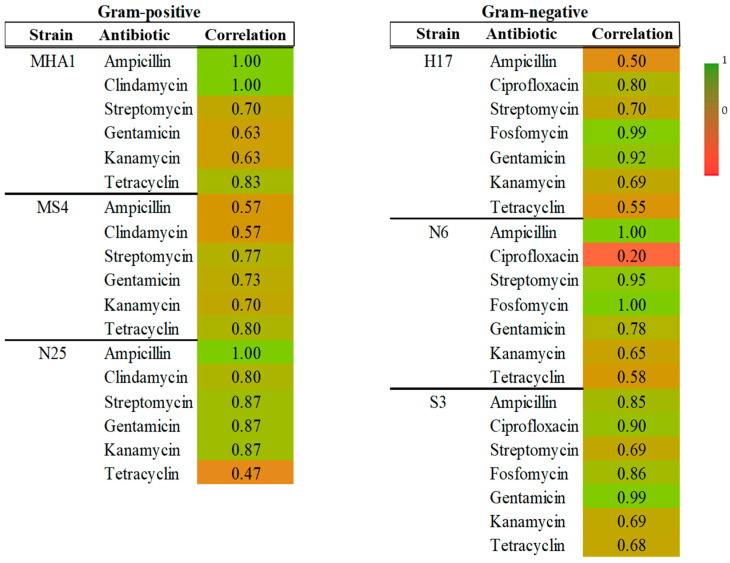
Heatmap showing correlation between MIC values and halo diameters for each strain and antibiotic in Gram-positive and Gram-negative groups.

**Figure 3 antibiotics-15-00014-f003:**
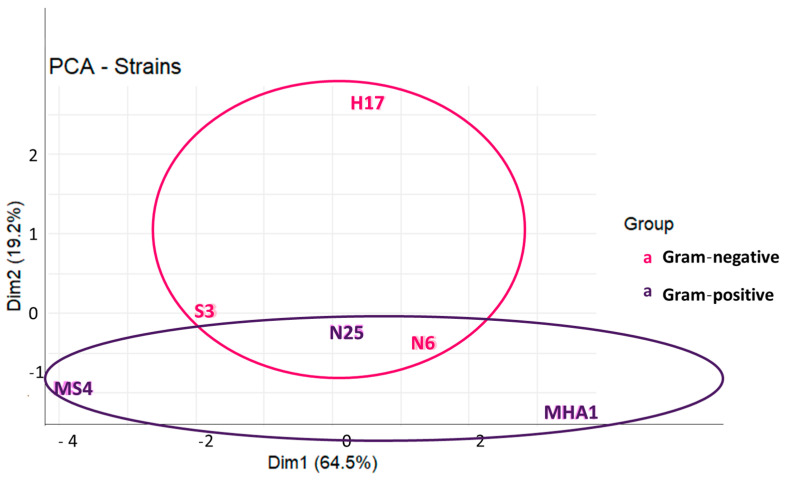
Principal components analysis performed on a dataset composed of the values of MIC and halo diameters for common antibiotics used in both Gram-positive and Gram-negative strains.

**Table 1 antibiotics-15-00014-t001:** Strains retrieved from the collection with the best plant growth promoting activities. Aux: production of auxins; PS: phosphate solublization; KS: potassium solubilization; Bio: formation of biofilms; N_2_-fix: nitrogen fixation; Sid: production of siderophores; ACC: ACC-deaminase activity; Amy: amylase activity; Cel: cellulase activity; Pect: pectinase activity; Prot: protesase activity; DNAse: DNAse activity; Chit: chitinase activity.

**Gram-Negatives**		
Strain	PGP properties and extracellular activities	Reference
H4	Aux, PS, KS, Sid, N_2_-fix, Amy, Prot	[22]
H17	KS, Sid, Aux, Bio, N_2_-fix, Prot	[22]
MH3	KS, Bio, Aux, Amy, Cel, Prot	[22]
N1	Sid, ACC, Bio, N_2_-fix	[23]
N3	PS, Sid, Aux, Bio, N_2_-fix, Cel, Chit	[23]
N6	Sid, Bio, N_2_-fix	[23]
N11	PS, Sid, Aux, Bio, N_2_-fix, Prot	[23]
N23	PS, Sid, Aux, Bio, N_2_-fix, Cel	[23]
N30	PS, Sid, Aux, ACC, N_2_-fix, Prot	[23]
S3	Bio, N_2_-fix, Aux, PS, KS, Sid, Cel	[21]
**Gram-Positives**		
Strain	PGP properties and extracellular activities	Reference
MS4	KS, Aux, N_2_-fix, Bio, Amy, Prot	[23]
MHA1	KS, Sid, Aux, N_2_-fix, Bio, Amy, Prot	[23]
MH6	Sid, Aux, N_2_-fix, Amy, Prot	[23]
MH9B	Aux, Sid, Cel, Prot	[23]
MR4	Bio, N_2_-fix, Sid, Aux, Prot, Amy, Pect, DNAase	[23]
MS5	KS, Sid, Aux, N_2_-fix, Amy, Prot	[21]
N14	PS, Sid, Aux, Bio, N_2_-fix, Prot	[23]
N18	PS, Sid, Aux, Bio, N_2_-fix, Prot, Pect, Cel	[23]
N25	PS, Sid, Aux, Bio, N_2_-fix, Prot, Cel, Chit	[23]
N32	PS, Sid, Aux, N_2_-fix, Prot, Pect, Cel	[23]

**Table 2 antibiotics-15-00014-t002:** Selected antibiotics for the determination of susceptibility or resistance in the bacterial samples.

Gram-Positives		Gram-Negatives	
Antibiotic	Amount on the Disk (micrograms)	Antibiotic	Amount on the Disk (micrograms)
Ampicillin	10	Ampicillin	10
Clindamycin	2	Ciprofloxacin	5
Chloramphenicol	30	Colistin sulphate	10
Erythromycin	15	Streptomycin	25
Streptomycin	25	Fosfomycin	50
Gentamicin	10	Gentamicin	10
Kanamycin	30	Kanamycin	30
Tetracycline	30	Tetracycline	30
Vancomycin	30		

**Table 3 antibiotics-15-00014-t003:** Inhibition zone diameters around antibiotic disks, measured in millimeters, for Gram-negative bacteria. Amp: Ampicillin; Cipro: Ciprofloxacin; Colis: Colistin; Fosfo; Fosfomycin; Genta: Gentamycin; Strepto: Streptomycin; Tetra: Tetracycline.

Strain	Amp	Cipro	Colis	Fosfo	Genta	Kana	Strepto	Tetra
H4	0	23	13	9	18	22	13	0
H17	0	0	0	19	21	19	0	17
MH3	9	24	11	12	26	27	20	11
N1	0	8	16	0	19	13	13	11
N3	9	30	0	37	19	11	18	14
N6	0	0	14	0	16	20	11	15
N11	0	20	12	16	19	19	15	10
N23	11	32	14	22	20	21	19	18
N30	12	31	10	23	21	0	21	0
S3	11	31	13	26	22	25	22	18

**Table 4 antibiotics-15-00014-t004:** Inhibition zone diameters around antibiotic disks, measured in millimeters, for Gram-positive bacteria. Amp: Ampicillin; Chloram: Chloramphenicol; Clinda: Clindamycin; Erythro: Erythromycin; Genta: Gentamycin; Kana; Kanamycin; Strepto: Streptomycin; Tetra: Tetracycline; Vanco: Vancomycin.

Strain	Amp	Chloram	Clinda	Erythro	Genta	Kana	Strepto	Tetra	Vanco
MHA1	0	0	0	0	17	17	9	8	0
MH6	15	29	14	39	29	33	26	27	7
MH9B	14	21	31	28	30	28	20	12	15
MR4	16	29	29	30	27	30	19	15	16
MS4	13	26	17	19	26	30	25	24	17
MS5	17	17	14	34	28	32	27	25	17
N14	14	27	13	29	25	29	23	16	24
N18	14	27	19	14	20	20	20	18	0
N25	0	17	6	12	19	14	16	14	8
**N32**	10	18	11	19	20	22	18	15	0

**Table 5 antibiotics-15-00014-t005:** Minimum inhibitory concentration (MIC) of selected Gram-positive bacteria for different antibiotics. In the lower row are breakpoints for Gram-positives when there are no data for the corresponding species according to EUCAST, 2024. n.d., there are no data. Amp: Ampicillin; Clinda: Clindamycin; Genta: Gentamycin; Kana; Kanamycin; Strepto: Streptomycin; Tetra: Tetracycline.

Strain	Amp (mg L^−1^)	Clinda (mg L^−1^)	Genta (mg L^−1^)	Kana (mg L^−1^)	Strepto (mg L^−1^)	Tetra (mg L^−1^)
MHA1	>512 (R)	>512 (R)	128	128	>512	256 (R)
MS4	>512 (R)	<0.5 (S)	8	4	8	<0.5 (S)
N25	>512 (R)	>512 (R)	16	8	32	2 (S)
Breakpoints for *Bacillus* spp. (except *B. anthracis*) [27]	n.d.	1	n.d.	n.d.	n.d.	n.d.
Breakpoints for Gram-positives when there are no other data [28]	0.5	0.5	n.d.	n.d.	n.d.	2

**Table 6 antibiotics-15-00014-t006:** Minimum inhibitory concentration of selected Gram-negative bacteria for different antibiotics. In the lower row are breakpoints for Gram-negatives when there are no data for the corresponding species according to EUCAST. n.d., there are no data. Amp: Ampicillin; Cipro: Ciprofloxacin; Fosfo; Fosfomycin; Genta: Gentamycin; Kana: Kanamycin; Strepto: Streptomycin; Tetra: Tetracycline.

Strain	Amp (mg L^−1^)	Cipro (mg L^−1^)	Fosfo (mg L^−1^)	Genta (mg L^−1^)	Kana (mg L^−1^)	Strepto (mg L^−1^)	Tetra (mg L^−1^)
H17	16 (R)	128 (R)	8 (S)	8 (R)	64 (S)	64	4 (R)
N6	>512 (R)	2 (R)	>512 (R)	64 (R)	64 (S)	64	4 (R)
S3	128 (R)	1 (R)	4 (S)	4 (S)	16 (S)	32	1 (S)
Breakpoints for *Pseudomonas* spp.	n.d.	0.5	64	4	64	n.d.	2
Breakpoints for Enterobacterales [27]	8	0.25	8	2	n.d.	n.d.	4
Breakpoints for Gram-negatives when there are no other data [28]	8	0.25	n.d.	n.d.	n.d.	n.d.	2

**Table 7 antibiotics-15-00014-t007:** Summary statistics of genome assemblies.

Genomic Features	MHA1	MS4	N25	H17	N6	S3
Total length (bp) ^a^	5,891,127	6,043,838	5,805,119	4,949,670	6,641,484	5,276,853
Number of contigs (≧100 bp) ^a^	200	219	92	74	72	170
Largest contig (bp) ^a^	1,407,233	1,236,212	842,001	1,093,983	1,263,707	1,548,608
N50 (bp) ^a^	1,005,957	1,008,741	330,999	404,426	412,829	414,608
L50 ^a^	3	3	5	3	5	3
N75 (bp) ^a^	227,051	228,602	211,237	288,132	235,369	229,884
L75 ^a^	6	6	11	7	10	8
DNA G + C content (mol%) ^a^	37.46	37.44	34.79	56.41	64.25	54.40
Ns per 100 kb ^a^	0	0	0	0	0	0
Mean coverage	74.97×	73.40×	78.81×	62.05×	47.05×	41.72×
Number of CDS ^b^	5958	6185	5729	4577	5963	4955
RNA genes ^b^	182	162	127	97	65	88
rRNA ^b^	27	27	19	14	4	13
tRNA ^b^	154	134	107	82	60	74
tmRNA ^b^	1	1	1	1	1	1
Number of contigs as plasmid ^c^	10	20	1	1	0	3
CRISPR repeat ^d^	0	7	5	3	5	2

^a^ Data from QUAST v. 5.3.0 software [29]. ^b^ Data from Prokka v.1.14.5 [30]. ^c^ Data from Deeplasmid software, version August 22, 2024 [31]. ^d^ Data from CRISPRCasFinder, version 4.2 [32].

**Table 8 antibiotics-15-00014-t008:** Identification of the six selected strains based on the sequence of the 16S rRNA gene sequence and dDDH. The strain S3 is identified by the bioproject since a percentage of dDDH below 70% could indicate a novel species, which is an aspect that is under study.

Strain	16S rRNA (bp)	Most Similar Taxon	Identity (%)	dDDH Value	Accession Number
N6	1527	*Achromobacter spanius*	100%	89.3% (*A. spanius*)	GCA_977062005
H17	1438	*Leclercia adecarboxylata*	99.71%	74.2% (*Leclercia tamurae*)	GCA_977061975
				44.8% (*L*. *adecarboxylata*)	
S3	1443	*Pantoea conspicua*	99.64%	69.8% (*Pantoea pleuroti*)	PRJEB102903
				29.7% (*P. conspicua*)	
N25	1318	*Bacillus cereus*	100%	72.5% (*B. cereus*)	GCA_977062015
MHA1	1459	*Priestia aryabhattai*	100%	73.1% (*P. aryabhattai*)	GCA_977061995
MS4	1560	*Priestia megaterium*	99.86%	92.0% (*P. megaterium*)	GCA_977061985

dDDH values ≥ 70% indicate same species. dDDH values < 70% indicate possible new species [34].

**Table 9 antibiotics-15-00014-t009:** Antimicrobial resistance genes found in the genomes of Gram-positive strains, number of copies, and localization. ARG: number of antibiotic resistance genes; N: total number of ARG found by PATRIC; n: number of genes found by PATRIC related to the antibiotics used in this work. Tet., Tetracycline; Strep., Streptomycin; Chloram., Chloramphenicol.

Strain	ARG N/n	Gene	Number of Copies	Localization	Confers Resistance Towards
MHA1	49/10	*gyrA, gyrB*	2 copies each	Chromosome	Ciprofloxacin
		*s10p*	1 copy	Chromosome	Tetracycline
		*s12p*	1 copy	Chromosome	Streptomicin
		*gidB*	1 copy	Plasmid	Streptomycin
		*murA*	1 copy	Chromosome	Fosfomycin
		*fosA*	1 copy	Chromosome	Fosfomycin
MS4	49/10	*gyrA, gyrB*	2 copies each	Chromosome	Ciprofloxacin
		*s10p*	1 copy	Chromosome	Tetracycline
		*s12p*	1 copy	Chromosome	Streptomicin
		*gidB*	1 copy	Chromosome	Streptomycin
		*murA*	2 copies	Chromosome	Fosfomycin
		*fosB*	1 copy	Chromosome	Fosfomycin
N25	52/17	*gyrA, gyrB*	1 copy each	Chromosome	Ciprofloxacin
		*s10p*	1 copy	Chromosome	Tetracycline
		*s12p*	1 copy	Chromosome	Streptomicin
		*gidB*	1 copy	Chromosome	Streptomycin
		*murA*	2 copies	Chromosome	Fosfomycin
		*fosB*	1 copy	Chromosome	Fosfomycin
		*BcII* family	1 copy	Chromosome	Ampicillin
		*CatA15/A16*	1 copy	Chromosome	Chloramphenicol
		*YkkA, YkkC, YkkD*	1 copy each	Chromosome	Tet., Strep., Chloram.
		*VanXY*-unclassifield	2 copies	1(chr) + 1(plas)	Vancomycin
		*VanA/I/Pt*-type	1 copy	Plasmid	Vancomycin
		*VanF/M*-type	2 copies	1(chr) + 1(plas)	Vancomycin
		*VanR*- unclassified	1 copy	Plasmid	Vancomycin

**Table 10 antibiotics-15-00014-t010:** Determinants for antimicrobial resistance found in Gram-negative strains, number of copies, and localization. ARG: number of antibiotic resistance genes; N: total number of ARG found by PATRIC; n: number of genes found by PATRIC related to the antibiotics used in this work. Amp., Ampicillin; Cipro., Ciprofloxacin; Chloram., Chloramphenicol.; Genta., Gentamicin; Tet., Tetracycline; (chr), chromosome; (plas), plasmid.

Strain	ARG (N/n)	Gene	Number of Copies	Localization	Confers Resistance Towards
H17	55/26	*BlaEC*	1 copy	Chromosome	Ampicillin
		*marA, marB*	1 copy each	Chromosome	Activator protein (regulation)
		*AcrA, AcrB, AcrD, AcrE,*	1 copy each	Chromosome	Amp., Cipro., Chloram., Genta., Tet.
		*AcrF, AcrZ*	1 copy each	Chromosome	Amp., Cipro., Chloram., Genta., Tet.
		*TolC*	1 copy	Chromosome	Export channel, several antibiotics
		*TolC/OmpH*	1 copy	Chromosome	Export channel, several antibiotics
		*MdtA, MdtB*	1 copy each	Chromosome	Multidrug efflux pump
		*MdtC*	2 copies	Chromosome	Multidrug efflux pump
		*MdtL*	1 copy	Chromosome	Multidrug efflux pump
		*H-NS*	1 copy	Chromosome	Hist.-like nucleoid-structur., Colistin Ciprofloxacin
		*gyrA, gyrB*	1 copy each	Chromosome	
		*MacA, MacB*	1 copy each	Chromosome	Erythromycin
		*s12p*	1 copy	Chromosome	Streptomycin
		*gidB*	1 copy	Chromosome	Streptomycin
		*murA*	1 copy	Chromosome	Fosfomycin
		*MdfA/Cmr*	1 copy	Chromosome	Multidrug efflux pump
		*EmrD*	1 copy	Chromosome	Chloramphenicol
N6	50/17	*MexAB-OprM*	2 copies	Chromosome	Multidrug efflux pump
		*MexCD-OprJ*	3 copies	Chromosome	Multidrug efflux pump
		*MexHI-OpmD*	3 copies	Chromosome	Multidrug efflux pump
		*OprM*	1 copy	Chromosome	Export channel, several antibiotics
		*gyrA, gyrB*	1 copy each	Chromosome	Ciprofloxacin
		*MacA, MacB*	1 copy each	Chromosome	Erythromycin
		*s12p*	1 copy	Chromosome	Streptomycin
		*gidB*	1 copy	Chromosome	Streptomycin
		*murA*	2 copies	Chromosome	Fosfomycin
S3	54/26	*MarA*	1 copy	Chromosome	Activator protein (regulation)
		*AcrA, AcrB, AcrD, AcrZ*	1 copy each	Chromosome	Multidrug efflux pump
		*TolC*	1 copy	Chromosome	Export channel, several antibiotics
		*MdtA, MdtB, MdtC*	2 copies each	Chromosome	Multidrug efflux pump
		*H-NS*	5 copies	3 (chr), 2 (plas)	Hist.-like nucleoid-structur., Colistin
		*gyrA, gyrB*	1 copy each	Chromosome	Ciprofloxacin
		*s12p*	1 copy	Chromosome	Streptomycin
		*gidB*	1 copy	Chromosome	Streptomycin
		*murA*	1 copy	Chromosome	Fosfomycin
		*MdfA/Cmr*	2 copies	Chromosome	Multidrug efflux pump
		*OprB*	1 copy	Chromosome	Export channel, several antibiotics
		*MexEF-OprN*	1 copy	Chromosome	Multidrug efflux pump

## Data Availability

All data are presented in the paper or as Appendix A. The sequences of the whole genomes of the strains have not been deposited in public repositories since they are being tested for commercial use with private companies. The sequences are available to interested researchers on request.

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
