# Peer review of "Antimicrobial Resistance in Bacterial Strains of Agricultural Interest: Predictions Based on Genomic Data"

_antibiotics, 2025, doi:10.3390/antibiotics15010014_

Round 1
Reviewer 1 Report
Comments and Suggestions for Authors
The manuscript raises an interesting issue concerning the existence of drug resistance among bacteria that can be used as plant growth promoters. However, in my opinion, the manuscript in its current form requires significant revision, as indicated below:
- the chapter on methodology should precede the description of the results (it would make it easier for the reader to analyze them). The numbering of the subsections should be corrected.
- the strains selected for the study should be characterized in more detail - why were these particular strains chosen, and perhaps it would be worth explaining whether they are in some way representative of the environment/possible applications (they origin from the laboratory collection - how are they representative for the real conditions?)
- I recommend removing Table 1 - in the absence of strain characteristics, it does not really contribute anything
- why was the “sum of halos” chosen as the criterion for selecting resistant and sensitive strains, and not, for example, the number of “ineffective” antibiotics in the case of a certain strain?
-lines 204-209 - this section requires further explanation as to the basis on which the authors draw this conclusion
- why these particular resistance genes were studied?
-lines 329-336 - this section is unclear and should be rewritten
- the discussion is extremely short and somewhat lacking in depth. It is more reminiscent of extensive conclusions. The conclusions, on the other hand, are general and only partially relate to the actual findings of the study.
Author Response
Reviewer 1
The manuscript raises an interesting issue concerning the existence of drug resistance among bacteria that can be used as plant growth promoters. However, in my opinion, the manuscript in its current form requires significant revision, as indicated below:
- the chapter on methodology should precede the description of the results (it would make it easier for the reader to analyze them). The numbering of the subsections should be corrected.
Thank you for your comment. We have strictly followed the instructions for authors and the template, and in this journal, the section Materials and Methods is situated after the Discussion.
- the strains selected for the study should be characterized in more detail - why were these particular strains chosen, and perhaps it would be worth explaining whether they are in some way representative of the environment/possible applications (they origin from the laboratory collection - how are they representative for the real conditions?)
Thank you very much for your comment. We have elaborated a new Table 1 with data about the strains. All these strains have been isolated from the rhizosphere of halophytes (plants able to thrive on saline soils) and have demonstrated good plant growth promoting activities. We selected 10 Gram positive and 10 Gram negative with good plant growth promoting activities. Some of them are able to solubilize phosphate or potassium, others fix nitrogen, other produce siderophores or auxins, and other display ACC-deaminase activity. Some of them display several properties and considered as the best PGPB. In Table 1, we disclose the activities of the strains.
- I recommend removing Table 1 - in the absence of strain characteristics, it does not really contribute anything
Please, see the precedent response. We amended Table 1.
- why was the “sum of halos” chosen as the criterion for selecting resistant and sensitive strains, and not, for example, the number of “ineffective” antibiotics in the case of a certain strain?
Since we had different behaviour of the strains to diverse antibiotics, we needed a criterion for selecting six strains to continue the work since we could not do the sequence analysis for all the strains due to budget restrictions. We decided to use the sum of all the halos, but we could also have used the maximum and minimum number of “zero” diameters, or the maximum number of ineffective antibiotics, as suggested by the reviewer. However, according to this comment and also as suggested by Reviewer #3 we have eliminated the sum of halos. We selected the strains with the smallest and largest halos for most of antibiotics.
-lines 204-209 - this section requires further explanation as to the basis on which the authors draw this conclusion
- why these particular resistance genes were studied?
The selection of the antibiotics used follows the criterion used by French regulation on biofertilizers. Neither in Spain nor in the whole European Community there is still a legislation about the levels of antibiotic resistance permitted in bacterial strains used in agriculture. However, we have received some requirements from French companies for evaluating the resistance towards antibiotics in agricultural strains and they always asked for these particular combinations of antibiotics for Gram-positives and Gram-negatives. Given the lack of regulation in Europe, we decided to use the French regulation. Once a common European regulation is stablished, similar analyses can be conducted with the stipulated antibiotics.
-lines 329-336 - this section is unclear and should be rewritten
Thank you for the comment. We have explained the paragraph in more depth, as follows:
In our study, we have multiple variables, such as 20 different bacteria (10 Gram-negative and 10 Gram positive), 5 common antibiotics for both types of bacteria, together with specific antibiotics for every of the bacterial types, determination of halo diameters, determination of MIC by two different methods (microdilution and agar dilution). In this regard, the multifactorial nature of the data can be simplified by reducing the number of variables to two principal components. In our case, both axes account for 83.7 % of the variability, with 64.5 % of variability associated to dimension 1 and 19.2 % of variability associated to dimension 2. Thus, the reduction of variables does not much affect the representativity of data and the conclusions. On the other hand, the study of principal component analysis (PCA) allows visual representation of the distance between samples and also allows to visualize how the samples can be grouped. We can see a distribution of the three Gram positive strains along the axis 1, whereas Gram negative strains get a central position, being distributed equally along both axes. This would suggest different resistance mechanisms in both types of bacteria, since the data of MIC and halos are grouped in a different way in Gram-positives and Gram-negatives. Besides, we can observe superposition of the circles englobing both types of bacteria, which could also suggest the existence of common resistance mechanisms.
- the discussion is extremely short and somewhat lacking in depth. It is more reminiscent of extensive conclusions. The conclusions, on the other hand, are general and only partially relate to the actual findings of the study.
Thank you very much for the comments. We have fully revised the discussion and do particular comments of each result in a deeper manner. We have introduced additional discussion on the localization and number of copies of the genes, according to Reviewer # 3. We have also introduced new references discussing the relevance and limitations of the study.
Reviewer 2 Report
Comments and Suggestions for Authors
Dear Authors, thank you for your interesting and meticulous work on such an important problem. Your publication will perfectly contribute to the topic of the special issue.
Please find my comments and questions below.
At least in my version of the manuscript, the resolution of Figure 1 is very low. Please check.
Please explain why not all tested bacteria in tables 3 and 4 were marked as sensitive or resistant?
Line 352, please, correct the spelling mistake.
Section 3.4, I suggest repositioning Tables 8 and 9 closer to the paragraph, where they are first mentioned.
Line 403: please change to italic font the species name Acinetobacter baumannii.
Please justify why the obtained sequences of the strains or their 16S rRNA gene sequences were not submitted to the database and no accession numbers are provided in this publication?
Has the whole genome shotgun sequencing provided the data required to identify the number of gene copies in the selected strains? If so, why were these data not evaluated, since you mention that it may be a reason for different susceptibility? Please, justify your decision or add this data.
Please explain and emphasize the novelty of your approach and findings. It must be brought out in the abstract or introduction. Because now it only states that you have chosen to investigate the strains of your collection. Please state a more ambitious goal, novelty, and value of the obtained results for the research on antibiotic resistance.
Author Response
Reviewer 2
Dear Authors, thank you for your interesting and meticulous work on such an important problem. Your publication will perfectly contribute to the topic of the special issue.
Please find my comments and questions below.
At least in my version of the manuscript, the resolution of Figure 1 is very low. Please check.
Thank you very much for the appreciation. We have changed the figure for a better resolution one.
Please explain why not all tested bacteria in tables 3 and 4 were marked as sensitive or resistant?
Thank you for the comment. Since the sum of the halos seems to be a criterion that can be misleading, we have deleted the sum of halos. Instead, we have explained the selection of the strains according to the smallest or largest halos for a high number of antibiotics. So we deleted any reference to “resistant” or “sensible” according to the halos since there is not an appropriated criterion for the species found in this work.
Line 352, please, correct the spelling mistake.
Thank you, we corrected it.
Section 3.4, I suggest repositioning Tables 8 and 9 closer to the paragraph, where they are first mentioned.
Thank you very much, we changed the positions of the Tables close to relevant text.
Line 403: please change to italic font the species name Acinetobacter baumannii.
Thank you, corrected.
Please justify why the obtained sequences of the strains or their 16S rRNA gene sequences were not submitted to the database and no accession numbers are provided in this publication?
Thank you very much. We have submitted the sequences of the 6 strains to database. We include the accession numbers in the revised version. For strain S3, Pantoea sp. S3, we presume that maybe it could be a novel species. In this regard, we used only the genus and we provide the bioproject number (we need to confirm whether it is a novel species or not).
Has the whole genome shotgun sequencing provided the data required to identify the number of gene copies in the selected strains? If so, why were these data not evaluated, since you mention that it may be a reason for different susceptibility? Please, justify your decision or add this data.
Thank you very much for the comment. We have done the assemblage of the whole genomes and we have identified the number of gene copies and the localization. We have included these data in Table 5 of the revised version,
Regarding some discrepancy between the number of genes and the resistance, this could be due to factors such as expression levels, specific regulation by environmental factors, or multiplicity of resistance by several synergic mechanisms. genes gene dose, expression and environmental regulation. This fact, of a similar number of genes in the “sensitive” and “resistant strains”. We have addressed these points in the discussion, which we have fully rewritten according to suggestions by reviewers 1 and 3.
Please explain and emphasize the novelty of your approach and findings. It must be brought out in the abstract or introduction. Because now it only states that you have chosen to investigate the strains of your collection. Please state a more ambitious goal, novelty, and value of the obtained results for the research on antibiotic resistance.
Thank you very much for the comment and the suggestion. We have included a section of Highlights and we have also emphasized the novelty and importance of the work for future research in the Discussion.
Reviewer 3 Report
Comments and Suggestions for Authors
Notes to the authors
The manuscript antimicrobial resistance in bacterial strains of agriculture interest: predictions based on genomic data, addresses an important topic-antimicrobial resistance in plant growth promoting bacteria- and combines phenotypic testing with whole-genome analysis.
However, despite these strengths, the study remains descriptive, lacks mechanistic depth, and does not provide significant scientific advancement beyond existing literature. Several methodological and interpretive issues limit the manuscript’s impact and suitability for publication in its current form.
The manuscript summarizes inhibition zones, MIC values, and genome-mined ARGs, but does not provide mechanistic insight, validation or genomic perspective. The important analysis is missing, such as plasmid vs chromosomal ARG localization, mobile genetic elements (integrons, transposons, insertion sequences), expression of resistance genes, efflux pumps, mutation analysis. Without these analyses, the study does not hold significant importance beyond previously published studies.
Antibiotic susceptibility was assigned using CLSI/EUCAST breakpoints for Bacillus, Pseudomonas, and Enterobacterales. This is not appropriate for PGRB and leads to questionable susceptible/resistant classifications, misinterpretation of MIC values, and inconsistent genotype-phenotype conclusions. This limitation affects the reliability of key results. The genome mining analysis lacks plasmid replicons, AMR operon integrity, mobility potential, contig structure, coverage of ARG hits. Without these inclusions, evaluation of ARG dissemination would be incomplete.
The authors only mention the discrepancies among the genotypes and phenotypes but failed to analyze and explore areas such as gene expression levels, efflux pumps, metabolic stress, environmental gene activation.
The discussion section repeats known AMR concepts but does not contextualize results. The authors state the AMR mechanisms that are already known. But they do not compare existing PGPB AMR data, compare genotype and phenotype mismatch among the strains, talk about ecological risk and regulatory implications. Summarizing inhibition zones does not capture the MDR patterns and may be misleading.
Minor comments
Introduction is overly long. Methods lack WGS assembly, statistics.
Figures are clear but may consider including error bars.
The manuscript must be toned for grammar and readability (especially the Introduction part)
Recommendation: Reject
The topic is important, and the dataset is promising, but the manuscript lacks the analytical, mechanistic depth and the methodological robustness required for publication. Major conceptual issues may be addressed by revision alone and thus I encourage the authors to expand the genomic analysis and incorporate functional validation for substantially improved future submission.
Comments on the Quality of English LanguageThe manuscript must be toned for grammar and readability (especially the Introduction part)
Author Response
Reviewer 3
The manuscript antimicrobial resistance in bacterial strains of agriculture interest: predictions based on genomic data, addresses an important topic-antimicrobial resistance in plant growth promoting bacteria- and combines phenotypic testing with whole-genome analysis.
However, despite these strengths, the study remains descriptive, lacks mechanistic depth, and does not provide significant scientific advancement beyond existing literature. Several methodological and interpretive issues limit the manuscript’s impact and suitability for publication in its current form.
Thank you very much for the comments. We have tried to palliate some deficiencies of the work and we are also aware of some of its limitations (for instance a wider number of strains and expression/regulatory studies), so we have included these limitations at the end of the discussion. However, we believe that our work can revindicate the importance of the subject and the necessity of a global and available database on antimicrobial resistance (halos and MICs) for the species/strains used at the agricultural and even environmental applications.
The manuscript summarizes inhibition zones, MIC values, and genome-mined ARGs, but does not provide mechanistic insight, validation or genomic perspective. The important analysis is missing, such as plasmid vs chromosomal ARG localization, mobile genetic elements (integrons, transposons, insertion sequences), expression of resistance genes, efflux pumps, mutation analysis. Without these analyses, the study does not hold significant importance beyond previously published studies.
Thank you very much for your comment. We have only done mining of the genes and tried to see correspondence with experimental data of halos and MICs. Our goal was also to see the possibility of predicting the existence of a high number of ARG in a particular soil, since this soil may be not adequate for isolating bacteria used as biofertilizers. Of course, the ARG localization in the genome (especially in mobile elements) is of the foremost importance. In this regard, we did the assemblage of the genomes and also looked at the localization of the genetic determinant in the chromosomes or in plasmids (please see Tables 9 and 10 in the revised version). No resistance genes were found in other mobile elements such as transposons, integrons, etc. at the time of dispersing these determinants in the environment. However, we did not do this kind of analysis, since we have not yet completed the assemblage of the whole genomes. In this work we just looked at the presence of the determinants in the genomes.
Antibiotic susceptibility was assigned using CLSI/EUCAST breakpoints for Bacillus, Pseudomonas, and Enterobacterales. This is not appropriate for PGRB and leads to questionable susceptible/resistant classifications, misinterpretation of MIC values, and inconsistent genotype-phenotype conclusions. This limitation affects the reliability of key results.
Thank you very much for the comment. In fact, this is one of the main conclusions of the work, i.e., the necessity of establishing a database for AMR in agronomical species/strains. The problem at the time of analysing the data is the comparison with clinical species or even with the table published by EUCAST “When there are not breakpoints” which is a general table for Gram negatives and Gram positives, but it is not customized for particular genera or species. We have clearly discuss this problem in the Discussion, in the conclusions and also in the Highlights: the necessity for a database based on agricultural species/strains
The genome mining analysis lacks plasmid replicons, AMR operon integrity, mobility potential, contig structure, coverage of ARG hits. Without these inclusions, evaluation of ARG dissemination would be incomplete.
Concerning the genetic mobility of the ARG, we agree in the necessity of exploring this aspect but we could not do plasmid replicons, AMR operon integrity, mobility potential, contig structure and coverage of ARG hits in this preliminary work in which we intend to call attention on the fact that AMR can be spread via biofertilizers and on the possibility of using genomics to predict such events, together with highlighting the lack of availability of experimental data of AMR in agricultural species.
The authors only mention the discrepancies among the genotypes and phenotypes but failed to analyze and explore areas such as gene expression levels, efflux pumps, metabolic stress, environmental gene activation.
Thank you very much for your comment. We also agree with the relevance of these studies. We did not approach transcriptomics analyses to see levels of gene expression, although we agree that the existence of environmental stress (for example, heavy metals) will increase the expression level. We have included comments to these points in the discussion.
The discussion section repeats known AMR concepts but does not contextualize results. The authors state the AMR mechanisms that are already known. But they do not compare existing PGPB AMR data, compare genotype and phenotype mismatch among the strains, talk about ecological risk and regulatory implications. Summarizing inhibition zones does not capture the MDR patterns and may be misleading.
Thank you for the comment. We have fully rewritten the discussion; we have compared our data with existing data of other studies of AMR in PGPB. These studies are not very frequent, but we have done a search to compare our results with previous data and we have included this discussion in the corresponding section.
In addition, we have discussed several possibilities to explain the discrepancy between the number of genes and the levels of resistance, including gene expression, regulation by environmental factors, synergy between several resistance mechanisms, etc.
Regarding the methodological approach of summarizing the halos of all the antibiotics together, we have deleted the sum of halos, since it could be misleading. We just selected the strains with the smallest and largest halos in general when considering all the antibiotics together.
Minor comments
Introduction is overly long. Methods lack WGS assembly, statistics.
The introduction has been revised and reduced. The raw sequence data of the strains is available with the accession numbers provided in the revised version.
Figures are clear but may consider including error bars.
Thank you very much. We included error bars in Figure 1. The data in Table correspond to MIC, for which three replications were done. In case of discrepancy (a value and the double of half), the highest MIC was considered in order not to artificially reduce the MICs.
The manuscript must be toned for grammar and readability (especially the Introduction part)
The introduction has been revised and reduced. The grammar and readability have been fully revised.
Recommendation: Reject
The topic is important, and the dataset is promising, but the manuscript lacks the analytical, mechanistic depth and the methodological robustness required for publication. Major conceptual issues may be addressed by revision alone and thus I encourage the authors to expand the genomic analysis and incorporate functional validation for substantially improved future submission.
Thank you very much for the comments. We have done the assemblage of the genomes and stablished the localization of the resistance determinants found in the genomes. In fact, the great majority of them were located in the chromosome and only two of them were found in plasmids. This could be taken as a relative guarantee of low probability of horizontal transfer by conjugation. Nevertheless, other HGT mechanisms can always operate, including transformation and transduction, although at much lower rate.
Regarding the functional validation, we have not performed expression analysis, influence of environmental conditions in the expression levels or mutagenesis studies. We plan to deep in our study and perform new analyses in the future.
Round 2
Reviewer 1 Report
Comments and Suggestions for Authors
The manuscript has been improved accordingly
Reviewer 3 Report
Comments and Suggestions for Authors
I appreciate the authors for incorporating the suggestions. The manuscript would be acceptable after screening the manuscript all over again for typos.
Comments on the Quality of English LanguageThe manuscript must be toned for grammar and readability (especially the Introduction part)